# Effect of an Al-TiO$_2$-C-Er$_2$O$_3$ Refining Agent on the Corrosion Resistance of a ZL205 Alloy

Ruiying Zhang [1,2,*], Yuqi Li [1], Jinxuan Li [1], Shuai Yang [1], Junhao Sha [1] and Zhiming Shi [1,2]

1. School of Materials and Engineering, Inner Mongolia University of Technology, Hohhot 010051, China; l1639108869@outlook.com (Y.L.); ljx1208118615@126.com (J.L.); yang_shuai888@126.com (S.Y.); l622301@126.com (J.S.); shizn@imut.cn (Z.S.)
2. Inner Mongolia Key Laboratory of Light Metal Materials, Hohhot 010051, China
* Correspondence: zhang_ruiying@126.com

**Abstract:** In this study, a new type of Al-TiO$_2$-C-Er$_2$O$_3$ refiner was prepared by exothermic dispersion, and the refiner was added to a ZL205 alloy to investigate the effect of the Al-TiO$_2$-C-Er$_2$O$_3$ refiner on the refinement effect and corrosion resistance of the ZL205 alloy. The results reveal that, with the increase in the Er$_2$O$_3$ content, the refining effect is first enhanced and then reduced. The best refining effect of the Al-TiO$_2$-C-Er$_2$O$_3$ refiner was observed by the addition of 6% Er$_2$O$_3$, which refined the grain size of the ZL205 alloy from 147 μm to 103.9 μm, and the grain size was 72% of the ZL205 alloy. The addition of the Al-TiO$_2$-C-Er$_2$O$_3$ refiner led to the improved bias agglomeration of the CuAl$_2$ phase at the grain boundaries in ZL205 and reduced the corrosion sensitivity of the alloy; moreover, the $\alpha$-Al nucleation temperature of the ZL205 alloy increased, the crystallization interval $\Delta T$ of the CuAl$_2$ phase increased from 7.4 °C to 9.3 °C, corresponding to an increase of 1.9 °C, and the self-corrosion current density (i$_{corr}$) of the ZL205 alloy was reduced from (2.83 × 10$^{-5}$ A/cm$^2$) to (6.26 × 10$^{-6}$ A/cm$^2$), a decrease of 77.8%; the polarization resistance (R$_1$ + R$_2$) was 1808.62 Ω·cm$^2$ in comparison with that of the original ZL205 alloy (480.42 Ω·cm$^2$), corresponding to a 276.7% increase.

**Keywords:** Al-TiO$_2$-C refining agent; ZL205; Er$_2$O$_3$; corrosion resistance

## 1. Introduction

As one of the most abundant metallic elements in the Earth's crust, aluminum is commonly used in automotive ships, electronic instruments, and other fields because of its good mechanical properties, machinability, and light-weight characteristics [1]. Cu is a key alloying element in aluminum alloys, and it exerts a certain solid solution strengthening effect. In addition, the CuAl$_2$ phase precipitated during aging exerts a significant aging strengthening effect, but the presence of the CuAl$_2$ phase increases the intergranular corrosion (IGC) tendency of Al–Cu alloys, thereby limiting the use of the alloy [2,3]. Yuanyuan Ji [4] et al. reported that the second phase in Al–Cu alloys precipitates on the grain boundaries and forms a potential difference, leading to increased corrosion. Aluminum–copper alloys exhibit different corrosion patterns because of their different use environments. According to the 1968 Dupont survey of metal material breakage in the United States, the loss produced by full-scale metal corrosion is as high as 31.5%, which is considerably greater than the damage produced by second stress corrosion [5]. Zhao [6] and Ajit Kumar Mishra [7] et al. reported that aluminum alloys are susceptible to some pitting corrosion in the inhomogeneous oxide films formed by non-metallic inclusions.

Several researchers have reported that the corrosion rate is reduced and that corrosion resistance is improved by the addition of rare earths [8]. T. M. Umarova [9] et al. investigated the effect of Ce on the corrosion resistance of Al–Fe alloys and also analyzed their corrosion behavior by the immersion of the alloy in a 3% NaCl solution. Their group found that the addition of Ce leads to the significant reduction in the self-corrosion potential and improvement in the corrosion resistance of the alloy. Zou [10] et al. investigated the effect

of rare-earth Yb on the organization and corrosion resistance of the Al–Si–Cu alloy. Their group found that the addition of Yb can generate an $Al_3Yb$ rare-earth phase in the alloy, which can form a protective film by attaching to the cathode Si surface after the corrosion of the alloy. That is, after the destruction of the alloy oxide film by $Cl^-$, the corrosion activity of the cathode is reduced; thus, the corrosion resistance of the alloy is improved. The effect of rare-earth elements on Al alloys has been gradually discovered and applied to production, which considerably improves product quality and production efficiency.

Grain refinement can reduce not only hot tearing and the cracking tendency of hot and direct cooling castings, but also segregation and porosity and improve surface finish and mechanical properties. A certain degree of grain refinement can improve the corrosion resistance of aluminum alloys [11]. Song Shenhua [12] and Liang Yanfeng [13] et al. investigated effects of refiners B, $TiB_2$, and $Al_3Ti$ on the corrosion of Al–Cu alloys and reported that the addition of refiners not only refines grains, but also prevents the precipitation of second-phase particles on the alloy grain boundaries, while the bias of enhanced particles toward the grain boundaries is a key reason for the increase in corrosion hangings; however, the excess of enhanced particles leads to the IGC of in situ materials. Liu Huan et al. [14] investigated the effects of rare-earth oxides on the refining agent: the average grain size of industrial pure aluminum is refined from 3800 μm to 320 μm with a good refinement effect, indicating that rare-earth oxides exert a significant effect on the refinement effect of the $Al\text{-}TiO_2\text{-}C$ refining agent. Therefore, in this paper, the effect of a new $Al\text{-}TiO_2\text{-}C\text{-}Er_2O_3$ refining agent on the refining effect and corrosion resistance of a ZL205 alloy by the addition of the rare-earth oxide $Er_2O_3$ into the $Al\text{-}TiO_2\text{-}C$ system refining agent is investigated.

## 2. Experimental Materials and Methods

First, a certain amount of the ZL205 alloy was added into a crucible resistance furnace, and the alloy was melted at 750 °C and warmed for 45 min. Second, 0.3 wt% of the $Al\text{-}TiO_2\text{-}C\text{-}XEr_2O_3$ refining agent was added and stirred well using a graphite rod. Next, the mixture was stirred with a graphite rod for 10 s and poured into a KBI standard ring mold preheated to 300 °C. Third, a half of the ZL205 alloy was polished step by step using sandpaper, followed by immersion in a 3.5% NaCl aqueous solution for 30 days to simulate the seawater corrosion test. After 30 days of corrosion, the specimen was removed and added into an 80 °C aqueous solution of $H_3PO_4$ and $Cr_2O_3$ (93% $H_2O$ + 5% $H_3PO_4$ + 2% $Cr_2O_3$) for 10 min after soaking in an alcohol solution for 3 min of ultrasonic cleaning to remove corrosion products. Scanning electron microscopy (SEM) was employed to observe the surface morphology after corrosion for the analysis of the effect of the refiner on the corrosion resistance of the ZL205 alloy. The specimens were anodized in a 2% $HBF_4$ solution at 25 V, 0.1 A, and 1–2 min by a polarized light microscope to observe the grain size, and the grain size was calculated by the intercept method for comparison. The alloy composition of ZL205 is shown in Table 1.

**Table 1.** Chemical composition of the ZL205 alloy (wt%).

| Cu | Mn | Ti | Cd | V | Fe | Si | Al |
|---|---|---|---|---|---|---|---|
| 4.6–5.3 | 0.3–0.5 | 0.15–0.35 | 0.15–0.25 | 0.05–0.30 | ≤0.15 | ≤0.06 | Residual content |

The thermal analysis experiment was conducted by temperature acquisition device DAQ-Central. The temperature measurement point was kept at the center of the crucible, and the measurement frequency was set to 0.005 times/s, so that both the phase change of the alloy could be investigated, and the beginning and end temperatures of the phase change and the duration of the phase change could be determined through the first-order differential curve.

For the electrochemical test, a Zahner-IM6e electrochemical workstation was used, the ZL205 alloy was used as the working electrode, and Pt was used as the auxiliary electrode because the auxiliary electrode cannot react with electrolyte solution and should

not be polarized or difficult to be polarized. The electrochemical test was carried out using the above-mentioned device to measure the dynamic potential polarization curve and electrochemical impedance spectrum of the specimen. The main equipment used in this test and its parameters are shown in Table 2.

**Table 2.** Main equipment of the experiment.

| Equipment Name | Equipment Model | Equipment Specifications |
| --- | --- | --- |
| Resistance Furnaces | SRJX-2-9 | $390 \times 320 \times 300$ mm<br>Maximum temperature 1000 °C |
| Microcomputer control electro-hydraulic servo Universal testing machine | SHT4605 | Accuracy level: 0.5<br>Maximum load: 600 KN |
| High-temperature test furnace | SXL-1700 | Furnace size: $400 \times 300 \times 300$ mm<br>rated temperature: 1650 °C |
| Olympus Optical Microscope | ZEISS | $50\times$, $100\times$, $200\times$<br>$500\times$, $1000\times$ |
| X-ray Diffractometer | D/MAX-2500/PC | Scanning Range: $-10°\sim+154°$ |
| Scanning Electron Microscope | FEG-650 | Magnification 60–1,000,000$\times$ |
| Electrochemical workstation | Zahner-IM6e | Frequency range: 10 µHz~3 MHz<br>Voltage Range: $\pm 10$ V<br>Current range: $\pm 0.1$ µA~$\pm 2$ A |

## 3. Analysis of Experimental Results

### 3.1. Refining Effect of the Al-TiO$_2$-C-Er$_2$O$_3$ Refining Agent on the ZL205 Alloy

Figure 1 shows the polarized photographs of the ZL205 alloy after the addition of the Al-TiO$_2$-C-XEr$_2$O$_3$ refining agent. By comparison, the alloy tissues were composed of equiaxed grains, but, with the increase in the Er$_2$O$_3$ content of the refining agent, the grains gradually became finer and smaller. By the addition of the Al-TiO$_2$-C-6%Er$_2$O$_3$ refining agent (Figure 1e), the ZL205 alloy exhibited the finest grains. Furthermore, with the continuous increase in the Er$_2$O$_3$ content of the refining agent (Figure 1f), the ZL205 alloy tissues did not exhibit any further refinement.

The grain size calculation reveals that the average grain size of the ZL205 alloy is ~147 µm (Figure 2). By using the Al-TiO$_2$-C refiner without Er$_2$O$_3$, the alloy was refined to about 127.2 µm, and the grain size was reduced by 12%. With the increase in the Er$_2$O$_3$ content of the refiner, the grain size of the alloy decreased significantly. By the addition of the Al-TiO$_2$-C-6% Er$_2$O$_3$ refiner, the grain size of the alloy decreased to 103.9 µm, and the grain size was 72% that of the ZL205 alloy without the refiner. Moreover, with the continuous increase in the Er$_2$O$_3$ content, the grain size did not change significantly. The results reveal that Er$_2$O$_3$ can improve the refinement effect of Al-TiO$_2$-C and that the optimum addition amount of Er$_2$O$_3$ in the Al-TiO$_2$-C-Er$_2$O$_3$ refinement agent is 6% under the test conditions.

After the Al-TiO$_2$-C-Er$_2$O$_3$ refiner enters the melt of the ZL205 alloy, the Al$_{20}$Ti$_2$Er phase and Al$_3$Ti phase dissolve and release the free [Ti] and Er atoms, which hinder the grain growth during solidification and improve the wettability of TiC particles, thus achieving the purpose of grain refinement. At the same time, the refiner can improve the distribution of CuAl$_2$ phase precipitated at the grain boundaries and reduce the CuAl$_2$ phase segregation phenomenon, which eventually improves the corrosion resistance of the alloy. Through the study of the refining effect of the refining agent, we found that, when the content of Er$_2$O$_3$ in the refining agent is too high, the refining effect is not significantly improved, but due to the refining agent, the presence of a large number of Al$_5$O$_{12}$Er$_3$ particles in the refiner increased the corrosion sites in the alloy and reduced the corrosion resistance of the alloy.

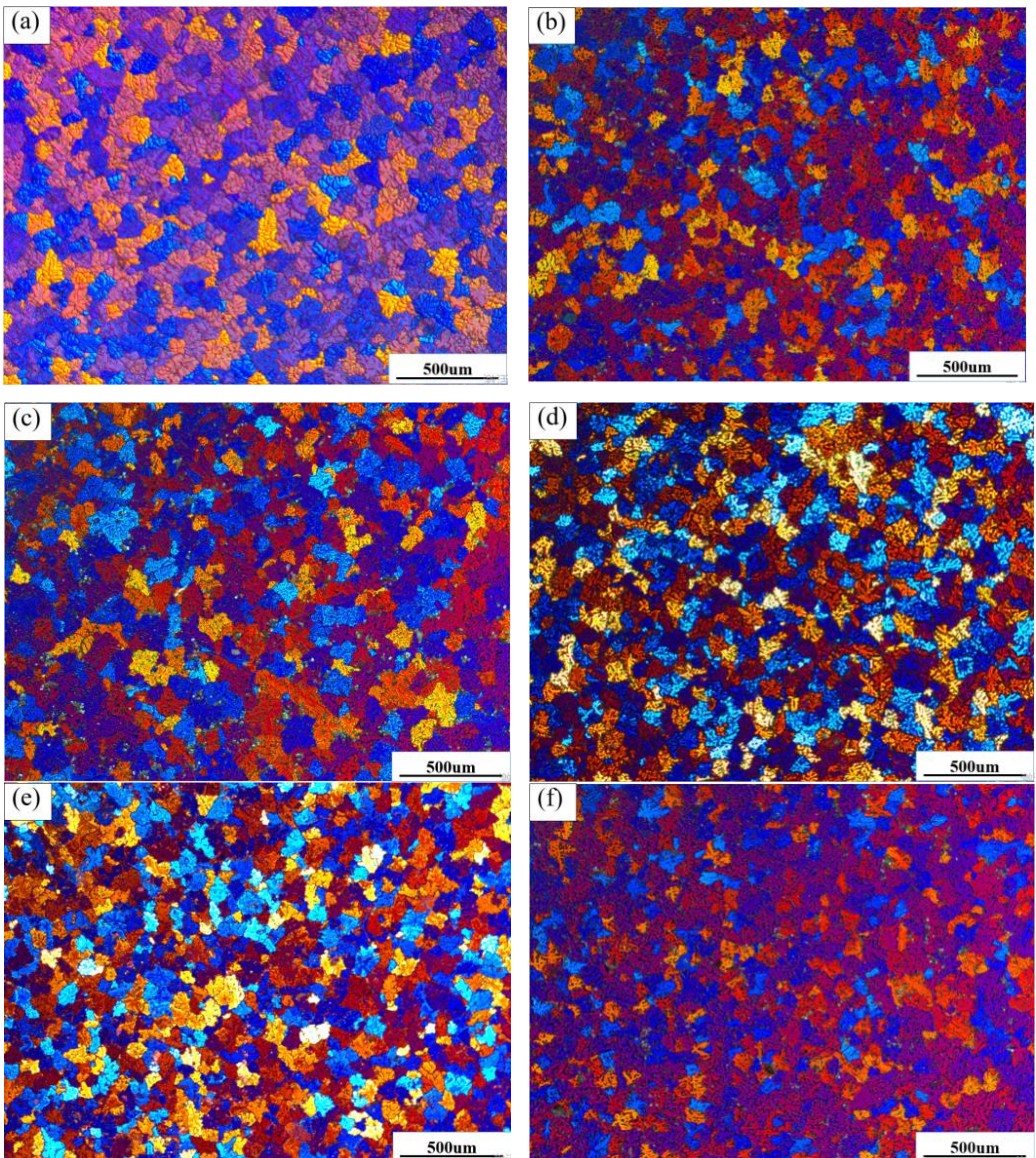

**Figure 1.** Polarized images of the ZL205 alloy after the addition of the 0.3 wt% Al-TiO$_2$-C-XEr$_2$O$_3$ refiner. (**a**) ZL205 alloy; (**b**) $X = 0$; (**c**) $X = 2$; (**d**) $X = 4$; (**e**) $X = 6$; (**f**) $X = 8$.

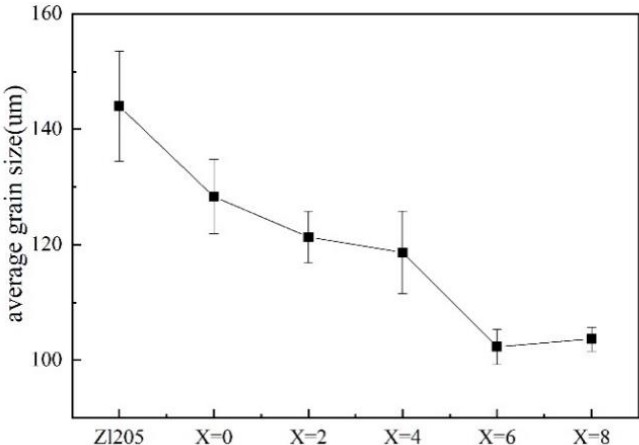

**Figure 2.** ZL205 alloy grain size after the addition of the 0.3 wt% Al-TiO$_2$-C-XEr$_2$O$_3$ refiner.

### 3.2. Thermal Analysis

Figure 3 shows the typical solidification curve and first-order differential curve of the ZL205 alloy: A clear turn in the first-order differential curve was observed, and corresponding exothermic peaks were generated. The first exothermic peak (point AB) corresponds to the crystallization process of α-Al. Point A corresponds to the initial nucleation temperature of α-Al, and point B corresponds to the nucleation growth temperature of α-Al. The second exothermic peak (point CD) corresponds to the crystallization process of the CuAl2 phase, and points C and D represent the initial nucleation temperature and growth temperature of the $CuAl_2$ phase, respectively. On the other hand, the temperature interval between the two points corresponds to the crystallization interval of the corresponding phase.

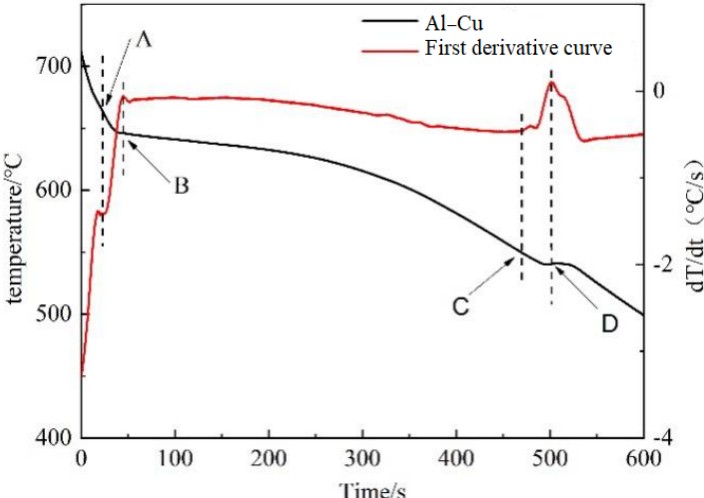

**Figure 3.** Typical ZL205 alloy solidification curve and first-order differential curve.

Figure 4 shows the thermal analysis curves of the ZL205 alloy with 0.3 wt% of different $Er_2O_3$ refiners, and Figure 4b shows the local enlargement: As the $Al_3Ti$ and TiC phases in the refiners can be used as the core of α-Al heterogeneous nucleation, the increase in the heterogeneous core reduced the supercooling degree required in the nucleation of the ZL205 alloy and improved the nucleation rate of the alloy. The release of a large amount of latent heat of crystallization during nucleation decreases the solidification rate of α-Al during solidification.

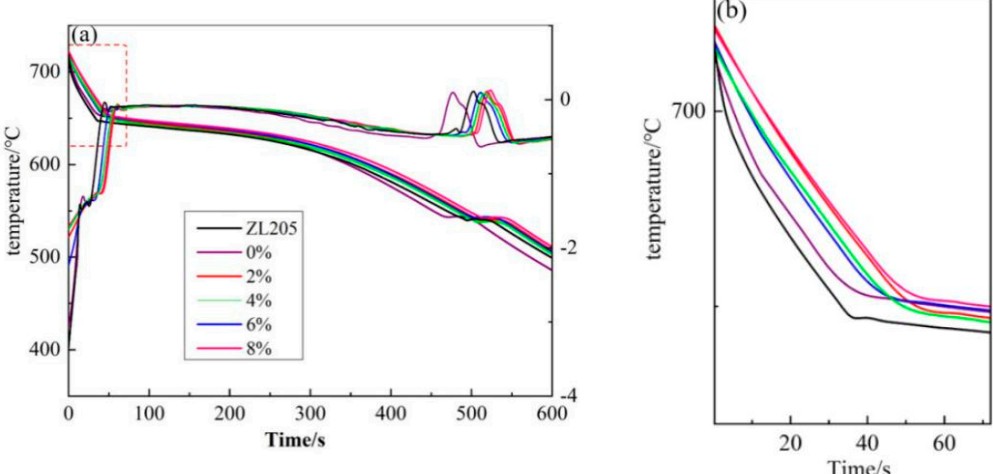

**Figure 4.** Thermal analysis curve of the ZL205 alloy with different erbium oxide contents. (**a**) Solidification curve and first-order differential curve. (**b**) Partial enlargement of area I.

Table 3 shows the solidification characteristic parameters of different specimens corresponding to the first-order differential curves. The initial nucleation temperature ($T_N^{\alpha\text{-Al}}$) of α-Al was 661 °C. By using the Al-TiO$_2$-C refiner without Er$_2$O$_3$, the initial nucleation temperature of α-Al was increased to 666.7 °C, thereby increasing the non-equilibrium liquid-phase line of the alloy. After the addition of the Al-TiO$_2$-C refiner containing Er$_2$O$_3$, the turning point in the first-order differential curve corresponding to the α-Al nucleation disappeared, indicating two rising curves with different slopes, because the rare-earth Al$_{20}$Ti$_2$Er phase generated free rare-earth Er at the reaction temperature [15], and the effect of rare-earth Er causes the initial nucleation process of α-Al to happen earlier and be longer in duration. However, due to the temperature selection problem, the exact nucleation temperature was not observed in the first-order differential curve.

**Table 3.** Solidification characteristic parameters of the sample.

| Sample | Phase Transformation Characteristic Temperatures | | | | |
|---|---|---|---|---|---|
| | $T_N^{\alpha\text{-Al}}$/(°C) | $T_G^{\text{Al}}$/(°C) | $T_N^{\text{Al-Cu}}$/(°C) | $T_G^{\text{Al-Cu}}$/(°C) | $\Delta T = T_N^{\text{Al-Cu}} - T_G^{\text{Al-Cu}}$ |
| ZL205 | 661.0 | 645.9 | 547.9 | 540.5 | 7.4 |
| Al-TiO$_2$-C | 666.7 | 646.2 | 551.0 | 543.3 | 7.7 |
| 2%Er$_2$O$_3$ | —— | 648.3 | 548.4 | 540.5 | 7.9 |
| 4%Er$_2$O$_3$ | —— | 651.1 | 552.0 | 543.5 | 8.5 |
| 6%Er$_2$O$_3$ | —— | 651.2 | 552.3 | 543.0 | 9.3 |
| 8%Er$_2$O$_3$ | —— | 651.5 | 552.1 | 542.8 | 9.3 |

After the addition of refiners, the eutectic CuAl$_2$-phase starting nucleation ($T_N^{\text{Al-Cu}}$) temperature increased overall, and the nucleation temperature reached the maximum value of 552.3 °C after the addition of the Al-TiO$_2$-C-6%Er$_2$O$_3$ refiner. The increase in the nucleation temperature reduced the subcooling required for nucleation. Studies have reported that the eutectic nucleation rate is positively correlated with subcooling [16] and that the alloy nucleation rate is low in the case of low subcooling. With the increase in the Er$_2$O$_3$ content of the refiner, the difference between the growth temperature of the CuAl$_2$ phase ($T_G^{\text{Al-Cu}}$) and the starting nucleation temperature $\Delta T$ increased significantly, and the crystallization interval △T of the ZL205 alloy was only 7.4 °C without the refiner (Table 3). By the addition of the Al-TiO$_2$-C refiner, the crystallization temperature of the CuAl$_2$ phase started to increase up to 543.3 °C, and the crystallization interval increased to 7.7 °C. With the increase in the Er$_2$O$_3$ content of the refiner, the crystallization interval further increased and reached the maximum value of 9.3 °C by the addition of 0.3 wt% of Al-TiO$_2$-C-6%Er$_2$O$_3$; this value is greater than that of the alloy without the refiner. Compared with the ZL205 alloy without refiner, the solidification temperature of CuAl$_2$ phase increased by 1.9 °C; after the continuous increase in the Er$_2$O$_3$ content, the interval did not grow continuously, probably because the refinement effect was not further improved by the addition of 8% Er$_2$O$_3$, while the content of Al$_5$Er$_3$O$_{12}$ impurity phase increased [17]. During the solidification process, the impurity phase segregates at the liquid-phase solidification front, hindering the generation of the CuAl$_2$ phase and causing a slight decrease in the nucleation temperature at the beginning of the CuAl$_2$ phase.

### 3.3. Effect of the Al-TiO$_2$-C-Er$_2$O$_3$ Refiners on the Corrosion Resistance of the ZL205 Alloy

Figure 5a–f show the surface morphology of the ZL205 alloy by the addition of the Al-TiO$_2$-C-XEr$_2$O$_3$ refiner after corrosion in a 3.5% NaCl solution for 30 days to remove the corrosion products. The CuAl$_2$ phase distributed along the grain boundary in the ZL205 alloy was shed during corrosion, and a small amount of pitting also was observed after corrosion. Relatively large corrosion pits were observed on the ZL205 alloy surface without the refiner, and a low number of pits were connected to form larger corrosion pits. A low number of pits were also found to be diffusely distributed around the matrix or grain boundaries. By the addition of the Al-TiO$_2$-C refiner, the number of pits on the ZL205

surface was significantly reduced, and a low number of granular $CuAl_2$-phase precipitates were observed at the grain boundaries, thereby significantly improving the corrosion resistance of the ZL205 alloy. By the addition of the $Al$-$TiO_2$-$C$-$6\%Er_2O_3$ refiner to the ZL205 alloy, fewer corrosion pits were observed (Figure 5e), IGC was significantly reduced, a high number of unexfoliated $CuAl_2$ phases were observed at the grain boundaries, and the depth of the pits was significantly lower than that of the ZL205 alloy without the refiner. Only marginal localized IGC was caused by the exfoliation of the $CuAl_2$ phase with the continuous increase in the $Er_2O_3$ content of the refiner (Figure 5f), and the number of corrosion pits compared with that of the ZL205 alloy after the addition of the $Al$-$TiO_2$-$C$-$6\%Er_2O_3$ refiner tended to increase, but the corrosion pits did not become larger, but converted to continuous IGC with a diffuse distribution in the matrix. The grain boundaries at the large continuous $CuAl_2$ phase disappeared, and only a low amount of the granular $CuAl_2$ phase was present. The corrosion hole was deeper than that of the alloy with a 6% $Er_2O_3$ refining agent, but, compared with that of the ZL205 alloy corrosion, it was shallow.

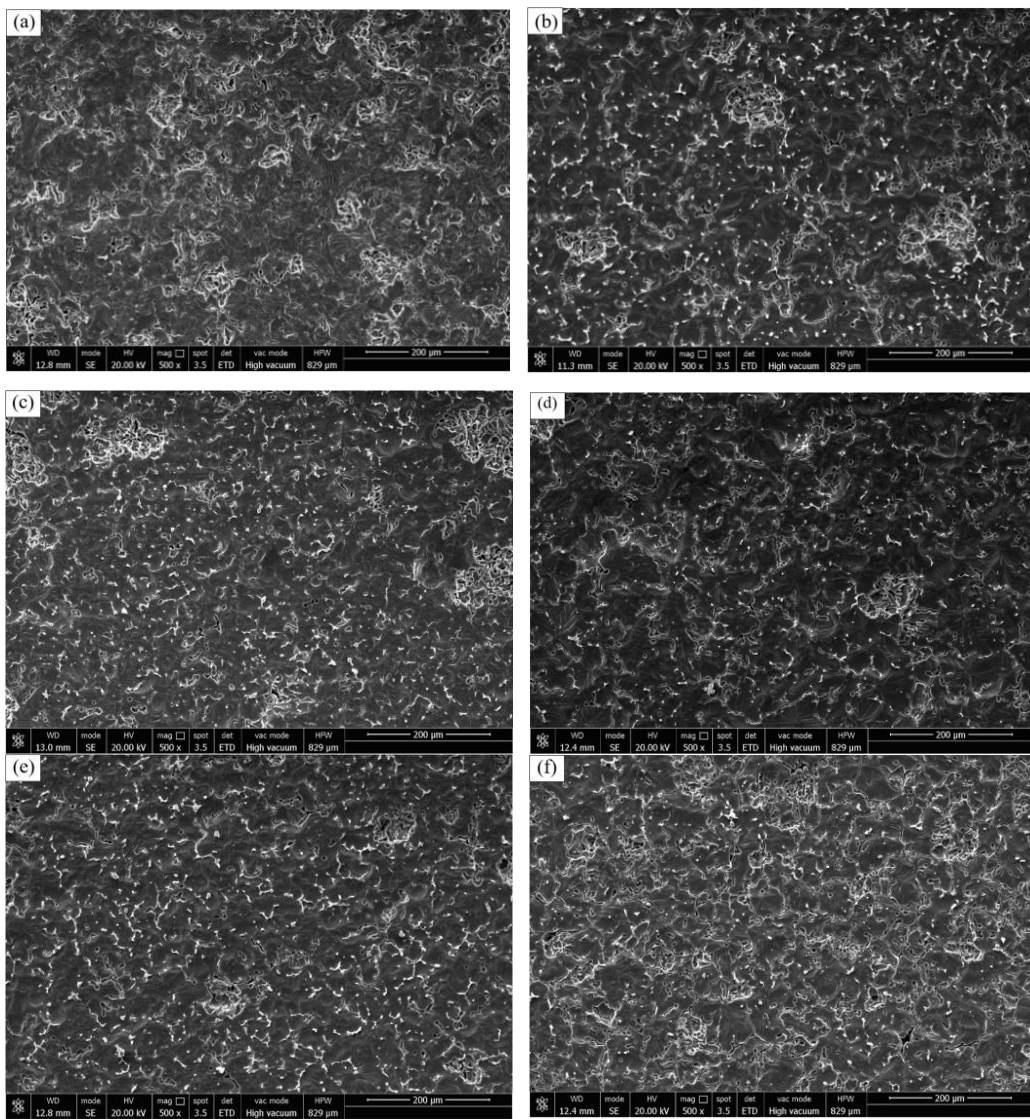

**Figure 5.** Surface morphology of the ZL205 alloy after 30-day corrosion after the addition of the $Al$-$TiO_2$-$C$-$XEr_2O_3$ refiner: (**a**) ZL205; (**b**) $X = 0$; (**c**) $X = 2$; (**d**) $X = 4$; (**e**) $X = 6$; (**f**) $X = 8$.

A typical reticulated CuAl$_2$ phase in the ZL205 alloy was observed as the eutectic reaction is not possible in alloys that are thermodynamically less than 5.8%, but in the ZL205 alloy under practical conditions, the critical Cu solubility and eutectic point also shifted due to the presence of other alloying elements (Mn, Ti, Cd, etc.) and non-stationary cooling (Figure 6a). As a result, the non-equilibrium eutectic CuAl$_2$ phases were formed, and these CuAl$_2$ phases precipitated at grain boundaries, thereby increasing the susceptibility of the alloy to IGC.

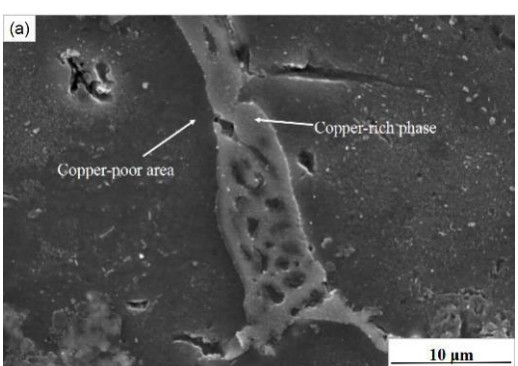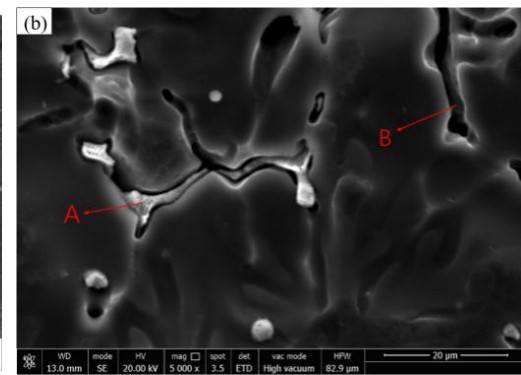

**Figure 6.** Micro-morphology of ZL205 and the surface morphology of the alloy after 30 days of corrosion after the addition of the refiner. (**a**) ZL205 alloy; (**b**) morphology after 30 days of corrosion in 3.5% NaCl.

IGC occurred mainly due to the low diffusion rate of Cu in aluminum solutions (Figure 6a), the net CuAl$_2$ phase precipitated at the grain boundaries corresponded to the Cu-rich zone, and on both sides, a Cu-poor zone was present. When an alloy is in a corrosive environment, a Cu-rich phase (CuAl$_2$ phase) in a 3.5% NaCl solution potential is generally $-0.61$ V, while the conventional potential of the 2-series aluminum alloy substrate is generally $-0.81$ to $-0.86$ V [18]; hence, there is a clear potential difference between the copper-rich area and its surrounding copper-poor area. In the local corrosion process, the copper-rich phase as the cathode and the surrounding copper-poor area constituted galvanic corrosion, leading to the anodic dissolution of the surrounding copper-poor area and the formation of IGC. Figure 6b shows the scanned image of the ZL205 alloy in the NaCl solution after 30 days of corrosion. The CuAl$_2$ phase in the A region was relatively intact after corrosion, and the surrounding Al matrix was corroded seriously, while for the B region, owing to severe corrosion, when the surrounding Al matrix corroded to a certain depth, the CuAl$_2$ phase was completely exposed, subsequently falling off after the cleaning of the corrosion product and eventually forming a long deep pit at the grain boundary. A small amount of the CuAl$_2$ phase was precipitated in the form of plasmas, and a similar corrosion occurred. Figure 7 shows the change in the morphology from the intermetallic compound to plasmas after corrosion, and Figure 7b–d shows the distribution of each element. It can be seen that there are obvious deep pits around the particles; the deep pits at the Al element are significantly reduced, and its corrosion starts from around the particles. There is an obvious potential difference between the Cu-poor area around the particles and the particles containing Cu elements, so that the Cu-poor area around the particles has corrosion priority; with the increase in corrosion time, corrosion gradually extends downward, and finally around the particles, the formation of craters occurs, as shown in the figure.

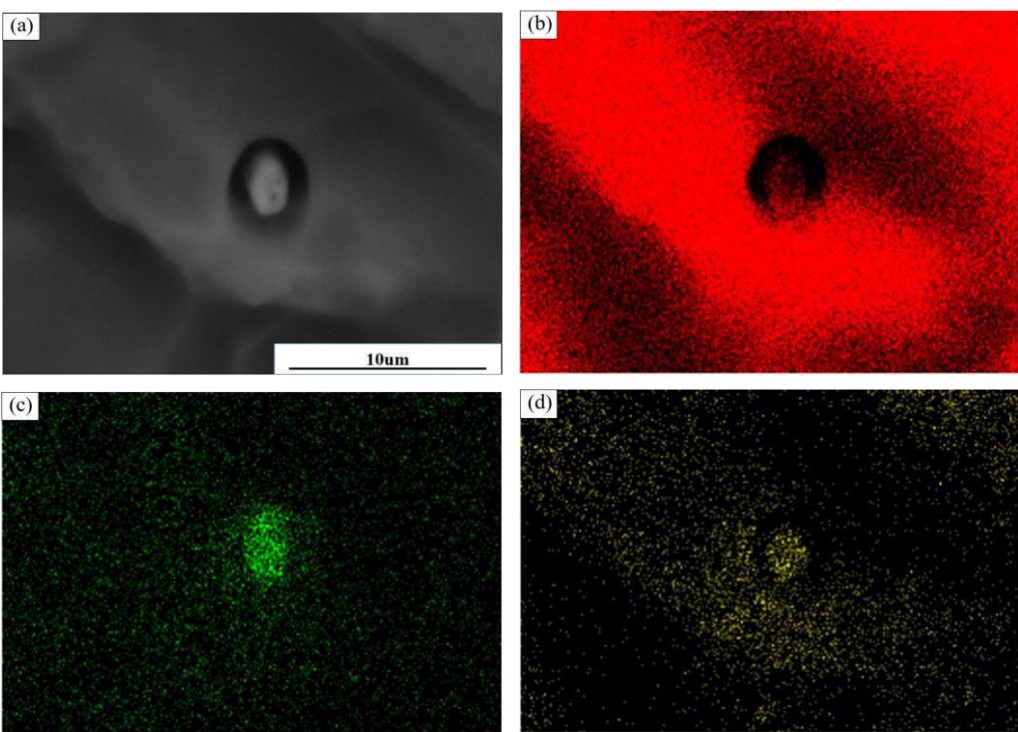

**Figure 7.** Scanning electron microscopy images of the corrosion profile of the ZL205 alloy. (**a**) Microstructure; (**b**) aluminum element; (**c**) copper element; (**d**) oxygen element.

In addition to being sensitive to IGC, the ZL205 alloy is also sensitive to pitting corrosion, which forms an un-dense oxide film layer on its surface. Meanwhile, the biased $CuAl_2$, second-phase particles, and some impurities affect the integrity of the oxide film, but the large amount of enriched $Cl^-$ destroys the passivation film of the material and eventually leads to pitting corrosion. The corrosion products mainly consist of $Al(OH)_3$. The reduction of $O_2$ and $H_2O$ mainly occurs at the cathode, while the dissolution of Al mainly occurs at the anode. The redox of oxygen occurs mainly as follows:

$$O_2 + 2H_2O + 4e^- \rightarrow 4OH^- \tag{1}$$

Owing to the destruction of the passivation film, the Al matrix around the particles was highly soluble in the corrosive environment, generating $Al^{3+}$ and eventually reacting with the $OH^-$ generated at the cathode to form an $Al(OH)_3$ precipitate. The reaction was as follows:

$$Al^{3+} + OH^- \rightarrow Al(OH)_3 \tag{2}$$

A large amount of $OH^-$ and $Al^{3+}$ generated $Al(OH)_3$ precipitates, which were attached on the outside of the etch pit, and these precipitates halted the exchange of material inside and outside the pit. A high amount of $Cl^-$ was present inside the pit, while $Al^{3+}$ hydrolysis generated a large number of $H^+$ to reduce the internal pH, resulting in the dissolution of the internal matrix Al; it eventually expanded the pit continuously, because pitting occurred mostly at the grain boundary precipitation. A part of the pitting pits connected together after a certain period of development or with IGC. These pitting pits converged together, the corrosion products were cleaned, and the grain boundary near the corrosion of the serious $CuAl_2$ phase and intergranular corrosion and the joint action of pitting corrosion were produced, as Figure 5a, in the larger corrosion pits.

### 3.4. Electrochemical Analysis of the ZL205 Alloy by the Al-TiO$_2$-C-XEr$_2$O$_3$ Refining Agent

Figure 8 shows the dynamic polarization curve of the ZL205 alloy in a 3.5% NaCl solution after the addition of the Al-TiO$_2$-C-XEr$_2$O$_3$ refiner: The shape of the polarization

curve was the same, and the polarization curve shifted to the left, indicating that the overall corrosion rate after the addition of the refiner tends to reduce the dynamic polarization curve parameters for the Tafel extrapolation simulation of the self-corrosion current. Table 4 summarizes the results of the self-corrosion current density ($i_{corr}$) and self-corrosion potential ($E_{corr}$). Compared with that of the original ZL205 alloy, the self-corrosion potential of the ZL205 alloy increased after the addition of the refiner because the refiner improved the porosity, segregation, and other defects of the ZL205 alloy and reduced its corrosion sensitivity. With the increase in the $Er_2O_3$ content of the $Al$-$TiO_2$-$C$-$XEr_2O_3$ refiner, the self-corrosion current density first reduced and then increased, reaching a minimum of $6.26 \times 10^{-5}$ A/cm$^2$ by the addition of the $Al$-$TiO_2$-$C$-$6\%Er_2O_3$ refiner; this value is about 77.8% less than that of the ZL205 alloy without the refiner ($2.83 \times 10^{-5}$ A/cm$^2$). This result indicated that the corrosion rate of the material in the 3.5% NaCl solution is slower and that the corrosion resistance is better because, when the $Er_2O_3$ content of the refining agent is 6%, the refining agent exerts the best effect on alloy refinement, and rare-earth Er plays a purifying role on the material; simultaneously, the $CuAl_2$ phase precipitated at the grain boundaries was evenly distributed, the number of active atoms then increased, the ability of the material surface passivation film generation increased, and the final material corrosion resistance was enhanced. Moreover, at an extremely high $Er_2O_3$ content of the refiner, the corrosion resistance of the alloy was reduced because of the excessive $Al_5O_{12}Er_3$ particles in the refiner, which increased the corrosion sites in the alloy.

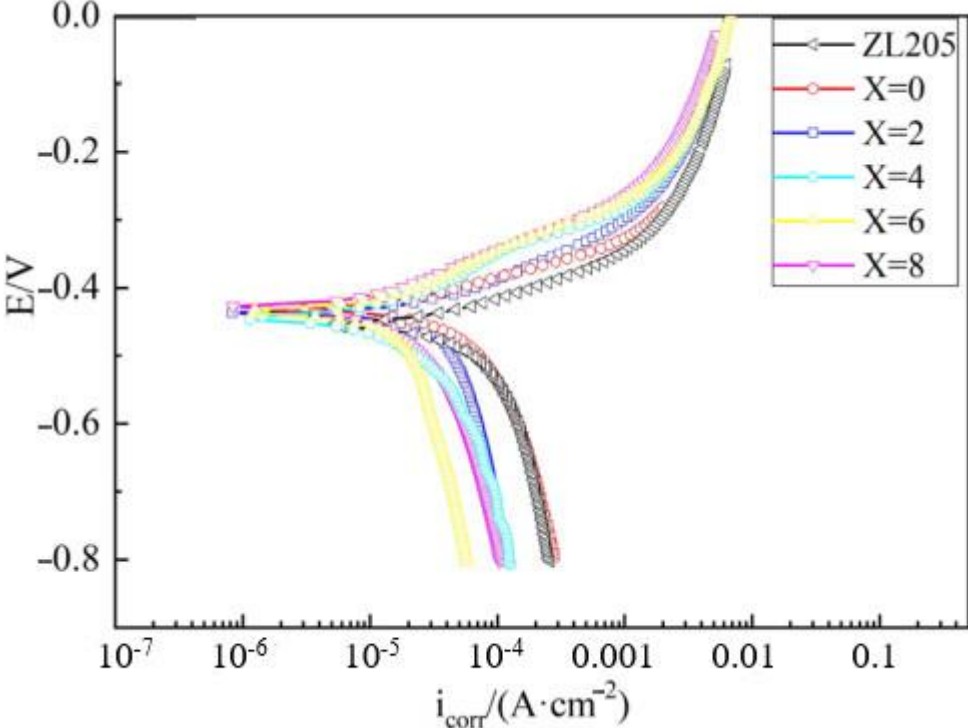

**Figure 8.** Potential polarization curve of the ZL205 alloy after the addition of the $Al$-$TiO_2$-$C$-$XEr_2O_3$ refiner.

**Table 4.** Electrochemical parameters of the ZL205 alloy after the addition of different $Er_2O_3$ refiners.

|  | ZL205 | *X* = 0 | *X* = 2 | *X* = 4 | *X* = 6 | *X* = 8 |
|---|---|---|---|---|---|---|
| $E_{corr}$/V | −0.452 | −0.432 | −0.437 | −0.442 | −0.437 | −0.427 |
| $i_{corr}$/(A·cm$^{-2}$) | $2.83 \times 10^{-5}$ | $1.97 \times 10^{-5}$ | $1.87 \times 10^{-5}$ | $6.72 \times 10^{-6}$ | $6.26 \times 10^{-6}$ | $7.21 \times 10^{-6}$ |

Figure 9 shows the Nyquist plot of the ZL205 alloy in a 3.5% NaCl solution: the Nyquist plots of all specimens revealed a high-frequency region of a capacitive resistance

arc, indicating that in the test process, the specimen and corrosion solution produced a charge transfer, and the generated passivation zone was not clear, which is consistent with the polarization curve measurement results. Figure 10 shows the fitting results of the EIS data for fitting: Without the addition of the refining agent, the ZL205 alloy exhibited the lowest polarization resistance, and with the increase in the $Er_2O_3$ content of the refining agent, the polarization resistance first increased and then decreased. In particular, at an $Er_2O_3$ content of 6% in the refining agent, the impedance was 1808.62 $\Omega \cdot cm^2$, and that of the original ZL205 alloy was 480.42 $\Omega \cdot cm^2$, corresponding to a difference of 376.7%. With the increase in the $Er_2O_3$ content to 8% in the $Al\text{-}TiO_2\text{-}C\text{-}Er_2O_3$ refiner, the impedance decreased significantly. A high impedance reflected that the solution current density is low; therefore, for the ZL205 alloy, with an $Er_2O_3$ content of 6% in the refiner, the alloy corrosion process exhibited the best inhibition effect, which was consistent with the polarization curve and corrosion morphology observation.

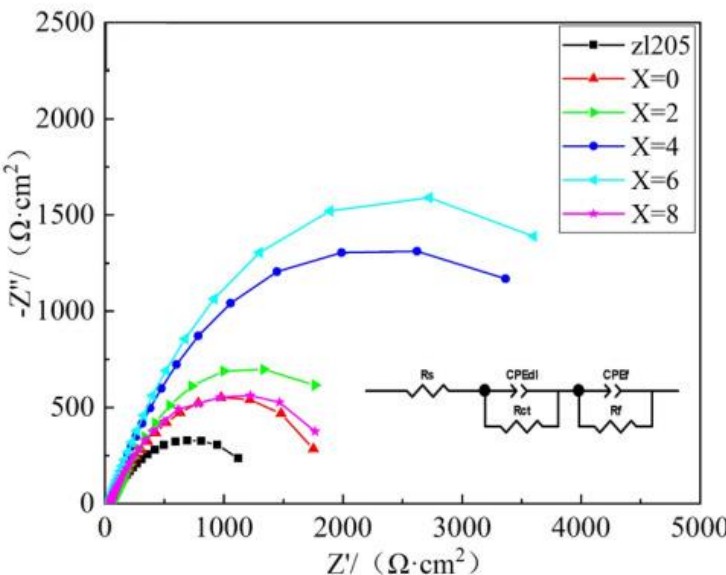

**Figure 9.** Nyquist diagram of the ZL205 alloy after the addition of the $Al\text{-}TiO_2\text{-}C\text{-}XEr_2O_3$ refiner.

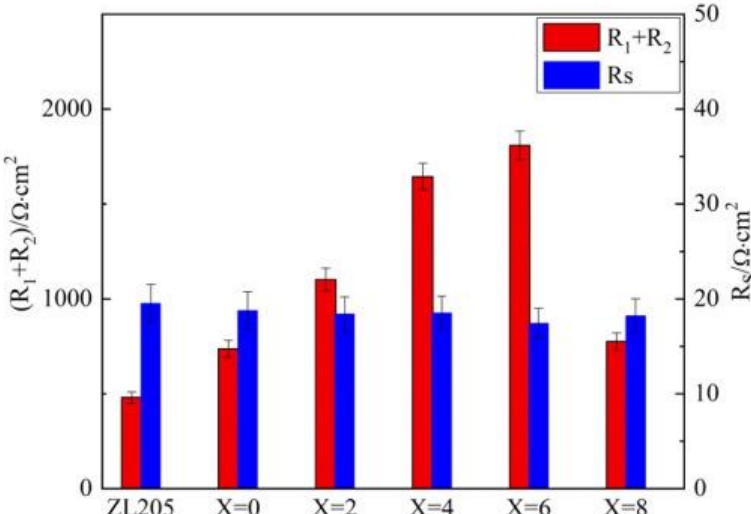

**Figure 10.** Comparison of the ($R_1 + R_2$) and Rs values in the equivalent circuit of the alloy in the as-cast state.

## 4. Conclusions

With the increase in the $Er_2O_3$ content, the refining effect of the Al-$TiO_2$-C-$Er_2O_3$ refiner on the ZL205 alloy was enhanced significantly, and at an $Er_2O_3$ content of 6%, the refining effect was the best, with a ZL205 alloy grain size refinement from 147 μm to 103.9 μm, and the grain refining size was 72% that of the ZL205 alloy; it improved α-Al in the ZL205 alloy. The initial nucleation temperature and crystallization interval of the $CuAl_2$ phase in the ZL205 alloy were increased, and the crystallization interval △T of the $CuAl_2$ phase increased from 7.4 °C to 9.3 °C, which was 1.9 °C higher.

The Al-$TiO_2$-C-$Er_2O_3$ refiner improved the bias agglomeration of the $CuAl_2$ phase at the grain boundaries and reduced the corrosion susceptibility of the alloy. The addition of the Al-$TiO_2$-C-$Er_2O_3$ refining agent led to the disappearance of the shedding phenomenon of large areas of the $CuAl_2$ phase at the ZL205 alloy grain boundaries; hence, corrosion resistance was significantly improved. The refining agent with a 6% $Er_2O_3$ content exhibited the best corrosion resistance effect, and the self-corrosion current density ($i_{corr}$) of the alloy was $6.26 \times 10^{-6}$ A/$cm^2$, and polarization resistance (R1 + R2) was 1808.62 Ω·$cm^2$, corresponding to 376.7% of the original ZL205 alloy impedance of 480.42 Ω·$cm^2$.

**Author Contributions:** R.Z., who is mainly responsible for the preparation of the first draft of the article; Y.L., mainly responsible for the post-article revision and touch-up; J.L., experimental operation and data measurement; S.Y., responsible for data processing, analysis and macro-regulation; J.S., responsible for the documentation of the experimental procedure.; Z.S., responsible for review and examination of the manuscript. All authors have read and agreed to the published version of the manuscript.

**Funding:** The present work was supported by the science and technology key project of Inner Mongolia (2021GG0252), Major science and technology projects of Inner Mongolia (2021ZD0030) and Inner Mongolia University of Technology Key Discipline Team Project of Materials Science (ZD202012).

**Institutional Review Board Statement:** Ethical approval was not required for this study.

**Informed Consent Statement:** Not applicable.

**Data Availability Statement:** Not applicable.

**Conflicts of Interest:** The authors declare no conflict of interest.

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
