# Peer review of "Effect of an Al-TiO2-C-Er2O3 Refining Agent on the Corrosion Resistance of a ZL205 Alloy"

_metals, doi:10.3390/met12061022_

Round 1

Reviewer 1 Report

Almost all writing was well designed. Figure 3-1 could be better in quality. From a metallurgical point of view, it would be better to explain more about the refining and its effect on electron path to pitting regions.

Author Response

Point 1: Almost all writing was well designed. Figure 3-1 could be better in quality. From a metallurgical point of view, it would be better to explain more about the refining and its effect on electron path to pitting regions.

Response 1: I have replaced the image in Figure 3-1 and here is my analysis from a metallurgical point of view.

After the Al-TiO2-C-Er2O3 refiner enters the melt of ZL205 alloy, the Al20Ti2Er phase and Al3Ti phase will dissolve and release the free [Ti] and Er atoms, which will hinder the grain growth during solidification and improve the wettability of TiC particles, thus achieving the purpose of grain refinement. At the same time, the refiner can improve the distribution of CuAl2 phase precipitated at the grain boundaries and reduce the CuAl2 phase segregation phenomenon, which eventually improves the corrosion resistance of the alloy. Through the study of the refining effect of the refining agent found that when the content of Er2O3 in the refining agent is too much, the refining effect is not significantly improved, but due to the refining agent The presence of a large number of Al5O12Er3 particles in the refiner increased the corrosion sites in the alloy and reduced the corrosion resistance of the alloy.(You can see it on lines 162-172, highlighting)

Reviewer 2 Report

  • The most important information which is missing is a detailed information about investigated alloy - what exactly the designation ZL205 means? According which standard this designation is presented? Authors should provide the exact chemical composition of the alloy.
  • Part 2 - Experimental. This part must be expanded. Authors should provide the information about all the experiments they conducted, also on which equipment etc. These are the basic rules for all the manuscripts and Authors should consider it.
  • Line 93 - what Authors mean by 'alloy tissues'? 
  • Line 167 - Authors wrote that 'CuAl2 phase distributed along the grain boundary'. On what basis Authors think so? There should be a proper microstructure characterization, e.g. observation on transmission electron microscope to analyze the second phase particles, also their location. In the present version of the manuscript it cannot be stated which precipiate/particles are present and where they are located.
  • The quality of the Figure 3-5 should be improved, or maybe the images should be at higher magnification in order to distinguish the type of corrosion. At the current quality and magnification it is difficult to say whether it is a IGC or uniform corrosion.
  • Line 212 - what Authors mean by the statement "A small amount of CuAl2 phase was precipitated in the form of plasmas"? What is a form 'plasmas'?
  • Line 259 - Authors wrote: "the polarization curve shifted to the left" - it is not true, which can be also seen by the parameters presented in Table 3-2. Please comment on that.
  • Line 262 - what Authors mean by self-corrosion current density (icorr) and self-corrosion potential (Ecorr)? In the literature it is used just corrosion current density and corrosion potential.
  • English language, but also grammar have to be improved. As an example the sentence, which starts in line 230 and ends in line 235 can be provided: "In addition to exhibiting serious sensitivity to IGC, the ZL205 alloy was also sensitive to pitting, and its surface generated an oxide film layer; however, the oxide film layer was not dense; simultaneously, the second-phase particles in the alloy and grain boundaries at the bias of the CuAl2 phase, as well as some impurities, affected the integrity of the oxide film, but a large number of enriched Cl destroyed the material itself passivation film equilibrium state". 
  • Point 3.4 - in the experimental part Authors did not write that potentiodynamic polarization experiments were conducted. These informations should be added.
  • There is a lack of any discussion in this manuscript. As an idea, maybe authors could provide a comparison of the effectivness of the addition of rare-earth elements on the grain refinement in comparison to other methods, such as severe plastic deformation.

Author Response

Point1:The most important information which is missing is a detailed information about investigated alloy - what exactly the designation ZL205 means? According which standard this designation is presented? Authors should provide the exact chemical composition of the alloy.

Response 1: ZL205 is a Chinese Al-Cu alloy, which is the earliest cast aluminum alloy used in industry. Its main performance characteristics are high mechanical properties at room temperature and high temperature, simple casting process, good cutting and machining performance, excellent heat resistance, and is the basis for the development of Cu-containing high-strength aluminum alloys and various heat-resistant alloys. The disadvantages are poor casting performance of solid-solution type alloy, large potential difference between copper-rich Al2Cu phase and Al matrix, low corrosion resistance and high density. The composition is shown in the table below.Add the table to line 88 of the article.

Table. Chemical compositions of ZL205 alloy(wt %)

Cu

Mn

Ti

Cd

V

Fe

Si

Al

4.6-5.3

0.3-0.5

0.15-0.35

0.15-0.25

0.05-0.30

≤0.15

≤0.06

Residual content

Point2:Part 2 - Experimental. This part must be expanded. Authors should provide the information about all the experiments they conducted, also on which equipment etc. These are the basic rules for all the manuscripts and Authors should consider it.

Response 2: (I have added Part2 part of the experimental procedure and its equipment, which you can see in lines 93-138 of the article.)

The thermal analysis experiment was conducted by temperature acquisition device DAQ-Central. The temperature measurement point is kept at the center of the crucible, and the measurement frequency is set to 0.005 times/second, so that the phase change of the alloy can be investigated, and the beginning and end temperatures of the phase change and the duration of the phase change can be determined through the first-order differential curve.

For electrochemical test, Zahner-IM6e electrochemical workstation is used, ZL205 alloy is used as the working electrode, and Pt is used as the auxiliary electrode because the auxiliary electrode cannot react with electrolyte solution and should not be polarized or difficult to be polarized. The electrochemical test is carried out by the above-mentioned device to measure the dynamic potential polarization curve and electrochemical impedance spectrum of the specimen.

Table. Main equipments of experiment

Equipment name

Equipment Model

Equipment specifications

Resistance Furnaces

SRJX-2-9

390×320×300mm

Maximum temperature 1000℃

Microcomputer control electro-hydraulic servo Universal testing machine

SHT4605

Accuracy level: 0.5

Maximum load: 600KN

High-temperature test furnace

SXL-1700

Furnace size: 400×300×300mm

 rated temperature: 1650°C

Olympus Optical Microscope

ZEISS

50X、100X、200X

500X、1000X

X-Ray Diffractometer

D/MAX-2500/PC

Scanning Range: -10°~+154°

Scanning Electron Microscope

FEG-650

Magnification 6×-1000000

Electrochemical workstation

Zahner-IM6e

Frequency range: 10μHz~3MHz

Voltage Range: ±10V

Current range: ±0.1μA~±2A

Point 3:Line 93 - what Authors mean by 'alloy tissues'? 

Response 3: The alloy tissues referred to here refers mainly to the α-Al matrix, with a small amount of Al2Cu phase and, of course, trace amounts of other phases.

Point 4: Line 167 - Authors wrote that 'CuAl2 phase distributed along the grain boundary'. On what basis Authors think so? There should be a proper microstructure characterization, e.g. observation on transmission electron microscope to analyze the second phase particles, also their location. In the present version of the manuscript it cannot be stated which precipiate/particles are present and where they are located.

Response 4: In Figure 3-6, the net CuAl2 phase precipitated at the grain boundary is the Cu-rich zone, and the Cu-poor zone is on both sides of it. Therefore, the copper-rich area (CuAl2 phase) and its surrounding copper-poor area there is an obvious potential difference, in the local corrosion process Copper-rich phase as the cathode and the surrounding copper-poor areas constitute galvanic corrosion, resulting in the surrounding copper-poor areas of anodic dissolution, the formation of intergranular corrosion.

Point 5: The quality of the Figure 3-5 should be improved, or maybe the images should be at higher magnification in order to distinguish the type of corrosion. At the current quality and magnification it is difficult to say whether it is a IGC or uniform corrosion.

Response 5We apologize for any errors in image quality. I have replaced the high quality images, For the differentiation of corrosion types, which can be seen in Figure 3-6(b) and illustrated in lines 210-216 of the article. Figure 3-6(b) shows the scanned image of the ZL205 alloy in the NaCl solution after 30 days of corrosion. The CuAl2 phase in the A region was relatively intact after corrosion, and the surrounding Al matrix was corroded seriously, while for the B region, owing to severe corrosion, when the surrounding Al matrix corroded to a certain depth, the CuAl2 phase was completely exposed, subsequently falling off after the cleaning of the corrosion product and eventually forming a long deep pit at the grain boundary. 

Point 6: Line 212 - what Authors mean by the statement "A small amount of CuAl2 phase was precipitated in the form of plasmas"? What is a form 'plasmas'?

Response 6: This should be the result of improper wording, and should be a small amount of CuAl2 phase will be precipitated in a point form.

Point 7: Line 259 - Authors wrote: "the polarization curve shifted to the left" - it is not true, which can be also seen by the parameters presented in Table 3-2. Please comment on that.

Response 7: Since the data is obtained from electrochemical workstation simulation, the picture is not clear enough. In this case the key data were intercepted to make Table 3-2, from which the leftward shift of the curve can be clearly seen.

Point8: Line 262 - what Authors mean by self-corrosion current density (icorr) and self-corrosion potential (Ecorr)? In the literature it is used just corrosion current density and corrosion potential.

Response 8: The dynamic polarization curve parameters were simulated by the Tafel extrapolation method to determine the corrosion rate. When the overpotential is large enough, the quantitative relationship between overpotential and current density is as follows:

η=a+blni

i is the current density; a, b are constants.

Point9: English language, but also grammar have to be improved. As an example the sentence, which starts in line 230 and ends in line 235 can be provided: "In addition to exhibiting serious sensitivity to IGC, the ZL205 alloy was also sensitive to pitting, and its surface generated an oxide film layer; however, the oxide film layer was not dense; simultaneously, the second-phase particles in the alloy and grain boundaries at the bias of the CuAl2 phase, as well as some impurities, affected the integrity of the oxide film, but a large number of enriched Cl- destroyed the material itself passivation film equilibrium state". 

Response 9: I have corrected the incorrect language and grammar in the article, thank you for your guidance.

Point10: Point 3.4 - in the experimental part Authors did not write that potentiodynamic polarization experiments were conducted. These informations should be added.

Response 10:(Here is my addition to the electrochemistry experiment section, which you can see in lines 99-104 of the article)

For electrochemical test, Zahner-IM6e electrochemical workstation is used, ZL205 alloy is used as the working electrode, and Pt is used as the auxiliary electrode because the auxiliary electrode cannot react with electrolyte solution and should not be polarized or difficult to be polarized. The electrochemical test is carried out by the above-mentioned device to measure the dynamic potential polarization curve and electrochemical impedance spectrum of the specimen.

Point11: There is a lack of any discussion in this manuscript. As an idea, maybe authors could provide a comparison of the effectivness of the addition of rare-earth elements on the grain refinement in comparison to other methods, such as severe plastic deformation.

Response 11: This is indeed the shortcoming of this paper, which only focuses on corrosion resistance and lacks the discussion of mechanical properties, etc. Grain refinement is the macroscopic property of rare earth refiners for aluminum alloys, while the discussion of corrosion resistance is the purpose of this experiment. Due to the special situation, it is impossible to make up some experiments in a short time, so the test of the properties is missing.

Round 2

Reviewer 2 Report

Thank you for all the answers. I recommend for the publication of the manuscript.